# Effect of Siponimod on Brain and Spinal Cord Imaging Markers of Neurodegeneration in the Theiler’s Murine Encephalomyelitis Virus Model of Demyelination

**DOI:** 10.3390/ijms241612990

**Published:** 2023-08-20

**Authors:** Suyog Pol, Ravendra Dhanraj, Anissa Taher, Mateo Crever, Taylor Charbonneau, Ferdinand Schweser, Michael Dwyer, Robert Zivadinov

**Affiliations:** 1Buffalo Neuroimaging Analysis Center, Department of Neurology, School of Medicine and Biomedical Sciences, University at Buffalo, State University of New York, Buffalo, NY 14203, USA; spol@bnac.net (S.P.); rdhanraj@bnac.net (R.D.); ataher89@yahoo.com (A.T.); mateocre@buffalo.edu (M.C.); tcharbonneau@bnac.net (T.C.); schweser@buffalo.edu (F.S.); mgdwyer@bnac.net (M.D.); 2Center for Biomedical Imaging, Clinical Translational Science Institute, University at Buffalo, State University of New York, Buffalo, NY 14203, USA

**Keywords:** TMEV, MRI, brain atrophy, spinal cord, siponimod

## Abstract

Siponimod (Sp) is a Sphingosine 1-phosphate (S1P) receptor modulator, and it suppresses S1P- mediated autoimmune lymphocyte transport and inflammation. Theiler’s murine encephalomyelitis virus (TMEV) infection mouse model of multiple sclerosis (MS) exhibits inflammation-driven acute and chronic phases, spinal cord lesions, brain and spinal cord atrophy, and white matter injury. The objective of the study was to investigate whether Sp treatment could attenuate inflammation-induced pathology in the TMEV model by inhibiting microglial activation and preventing the atrophy of central nervous tissue associated with neurodegeneration. Clinical disability score (CDS), body weight (BW), and rotarod retention time measures were used to assess Sp’s impact on neurodegeneration and disease progression in 4 study groups of 102 animals, including 44 Sp-treated (SpT), 44 vehicle-treated, 6 saline-injected, and 8 age-matched healthy controls (HC). Next, 58 (22 SpT, 22 vehicle, 6 saline injected, and 8 HC) out of the 102 animals were further evaluated to assess the effect of Sp on brain region-specific and spinal cord volume changes, as well as microglial activation. Sp increased CDS and decreased BW and rotarod retention time in TMEV mice, but did not significantly affect most brain region volumes, except for lateral ventricle volume. Sp suppressed ventricular enlargement, suggesting reduced TMEV-induced inflammation in LV. No significant differences in spine volume changes were observed between Sp- and vehicle-treated animals, but there were differences between HC and TMEV groups, indicating TMEV-induced inflammation contributed to increased spine volume. Spine histology revealed no significant microglial density differences between groups in gray matter, but HC animals had higher type 1 morphology and lower type 2 morphology percentages in gray and white matter regions. This suggests that Sp did not significantly affect microglial density but may have modulated neuroinflammation in the spinal cord. Sp may have some effects on neuroinflammation and ventricular enlargement. However, it did not demonstrate a significant impact on neurodegeneration, spinal volume, or lesion volume in the TMEV mouse model. Further investigation is required to fully understand Sp’s effect on microglial activation and its relevance to the pathophysiology of MS. The differences between the current study and previous research using other MS models, such as EAE, highlight the differences in pathological processes in these two disease models.

## 1. Introduction

In addition to the role of the autoimmune processes, we now have an increased understanding of the alternative inflammatory pathological cellular processes that drive neurodegeneration in individuals with multiple sclerosis (MS). The early stages of MS present autoimmune cell-driven demyelination, which triggers neurodegeneration, subsequently inducing secondary pathogenic mechanisms [1]. These cell death- and myelin loss-driven secondary pathogenic mechanisms and their complex interplay further accentuate the neurodegenerative process [2,3]. It is also suggested that neurodegeneration itself can lead to chronic MS pathologies, independent of immune cells, through distinct signaling pathways. This understanding has paved the way for novel therapeutic strategies that simultaneously target multiple mechanisms of neurodegeneration and have been shown to effectively treat patients in the progressive stages of the disease [3]. Among these strategies, the modulation of Sphingosine 1-phosphate (S1P) receptors has emerged as a promising approach.

S1P modulation, using drugs such as fingolimod, ponesimod, ozanimod, and siponimod (Sp), limits lymphocyte transport and suppresses autoimmune response-induced pathology. Out of the five S1P subtypes, Sp is a highly selective S1P inhibitor that targets subtypes 1 and 5 [4]. Furthermore, Sp has higher blood–brain barrier permeability in comparison to alternative S1P inhibitors. In additional to peripheral receptors, Sp can target S1P receptors expressed in CNS cell types such as microglia, astrocytes, and oligodendrocytes, in addition to lymphocytes [5,6].

In phase 3 EXPAND trials, Sp met its primary endpoint of reducing the risk of three-month confirmed disability progression, and showed a reduction in the percentage change in brain volume and change from baseline in the volume of T2 lesions [7]. It is hypothesized that these effects are driven in part by suppression of secondary neurodegenerative pathways in MS patients treated with Sp [8].

In preclinical testing, Sp treatment of experimental autoimmune encephalomyelitis (EAE) reduced astrogliosis and microgliosis; additionally, reduced lymphocyte infiltration in the St of EAE mice treated with Sp was found [9,10]. These results indicate that Sp has neuroprotective effects in the CNS of EAE mice. However, it is unclear if Sp’s EAE treatment effects are independent of Sp’s direct suppression of peripheral autoimmune response [11]. Hence, the current study aimed to characterize the effect of Sp on an animal model presenting inflammation-driven chronic demyelination and non-autoimmune cell-driven inflammatory pathways [12,13].

To this end, the study employed the Theiler’s murine encephalomyelitis virus (TMEV) model instead of EAE. While the EAE model presents acute autoimmunity-associated pathology, it does not present chronic demyelination nor non-immune driven inflammatory pathways. TMEV infection’s [12] disease course is closely linked to demyelinating lesions in the spinal cord, neuronal degeneration, and tissue loss [10]. TMEV-infected animals also exhibit gray matter (GM) atrophy, manifesting as a reduction in brain volume. This loss in volume is strongly correlated with clinical disability and a decline in motor skills observed in the infected animals [14]. Hence, TMEV is well-suited for testing the neuroprotective effects of Sp on MS pathology, which are driven by its action on non-immune cells expressing S1P receptors.

In this study, we characterized the effect of Sp on TMEV infection-driven neurodegeneration through high-field 9.4T MRI and histological changes within the spinal cord tissue. Treated and untreated TMEV animals’ brains and spinal cords, as well as controls’, were scanned over a 7-month period to evaluate the longitudinal volumetric changes in the different GM regions within the brain and lesion loads within the spinal cord tissue. Additionally, to characterize Sp’s effect at a functional level, the study animals were clinically evaluated for disability and motor skills.

## 2. Results

### 2.1. Siponimod Treatment Worsened Clinical Measures in TMEV-Induced Animals

#### Clinical Disability Score

The average CDS in the control groups, i.e., HC animals and SHAM animals, was similar (*p* = 0.714, two-way ANOVA). However, longitudinal changes in CDS between SpT and VT animal groups (*p* = 7.72 × 10^−13^, two-way ANOVA) were significantly different. Both TMEV groups had longitudinally significantly higher average CDS values compared to the HC (*p* < 0.001 vs. SpT group and *p* < 0.001 vs. VT group, two-way ANOVA) and SHAM (*p* < 0.001 vs. SpT group and *p* < 0.001 vs. VT group, two-way ANOVA) groups in the study (Figure 1A).

There was no significant difference between average CDS measures in SpT vs. VT animals from 10 wkPI up to 17 wkPI. Starting at 18 wkPI, SpT animals had significantly (*p* = 0.05, Welch’s ranked *T*-test, BH-adjusted) higher CDS (2.46 ± 1.2, mean ± SD, n = 22) compared to VT animals (2.24 ± 1.3, mean ± SD, n = 22). This difference remained significantly higher until the final timepoint, at 29 wkPI, for SpT (3.35 ± 0.67, mean ± SD, n = 22) relative to VT (2.88 ± 0.086, mean ± SD, n = 22) animals (Figure 1A).

### 2.2. Body Weight Measures

There was no longitudinal statistical difference in average BW measured between the control groups i.e., HC and SHAM animal groups (*p* = 0.994, two-way ANOVA), whereas the longitudinal changes within the TMEV groups, i.e., SpT and VT animal groups, were statistically different (*p* = 2 × 10^−16^, two-way ANOVA). Both TMEV groups had longitudinally significantly lower average BW in comparison to the control groups HC (*p* < 0.001 vs. SpT group and *p* < 0.001 vs. vehicle group, two-way ANOVA) and SHAM (*p* < 0.001 vs. SpT group and *p* < 0.001 vs. VT, two-way ANOVA) groups in the study (Figure 1B).

There was no significant difference between average BW measured within the SpT or VT TMEV groups up to 18 wkPI timepoints. At 19 wkPI, there was a significant difference (*p* = 0.04, Tukey’s test, BH-corrected) for the SpT (17.91 ± 1.98, mean ± SD g, n = 22) versus VT (18.87 ± 1.64, mean ± SD g, n = 22) animal groups comparisons. This difference remained significantly lower (*p* = 4.72 × 10^−4^, Tukey’s test, BH-corrected) up to the final timepoint at 29 wkPI, for SpT (17.74 ± 1.79, mean ± SD g, n = 22) relative to VT (18.94 ± 2.22, mean ± SD g, n = 22) animals (Figure 1B).

### 2.3. RotaRod Retention-Time

There was no significant difference in average Rotarod retention time observed between the control groups, i.e., HC and SHAM animal groups (*p* = 0.487, two-way ANOVA). However, the longitudinal changes within the TMEV groups, i.e., SpT and VT animal groups, were significantly different (*p* = 0.00139, two-way ANOVA). Both TMEV groups had longitudinally significantly lower average retention time compared to the control groups HC (*p* < 0.001 relative to SpT group and *p* < 0.001 vs. VT group, two-way ANOVA) and SHAM (*p* < 0.001 vs. SpT group and *p* <0.001 vs. VT group, two-way ANOVA) in the study (Figure 1C).

At 20 wkPI, there was a significant difference (*p* = 0.03, Tukey’s test, BH-corrected) between SpT (32.49 ± 31.99, mean ± SD seconds, n = 22) and VT (50.49 ± 36.20, mean ± SD seconds, n = 22) animal groups. This difference remained significantly lower (*p* = 0.01, Tukey’s test, BH-corrected) at the final timepoint at 28 wkPI for SpT (22.34 ± 26.32, mean ± SD seconds, n = 22) relative to vehicle (44.85 ± 35.28, mean ± SD seconds, n = 22) animals (Figure 1C).

### 2.4. Effects of Siponimod Treatment on TMEV Disease-Induced Changes on Brain Region-Specific Volumes

There were no significant differences for Cb and CC volumes between all treatment arms and control animal groups’ comparisons, neither for longitudinal comparisons nor at any individual timepoint (Figure 2C,D).

### 2.5. Anterior Commissure

For the WM region of measured AC volume, there were no significant differences for longitudinal changes in SpT animals compared to VT animals (*p* = 0.64, two-way ANOVA). There was a significant longitudinal change difference for the volume measured in SpT animals (*p* = 0.002, two-way ANOVA) and VT animals (*p* = 0.001, two-way ANOVA) compared to the SHAM control animal group. At the timepoint 12 wkPI, there was a significant difference (*p* = 0.04, Tukey’s test, BH-corrected) between the SpT animals (0.41 ± 0.05, mean ± SD mm^3^, n = 11) compared to HC animal groups (0.35 ± 0.03, mean ± SD mm^3^, n = 3) at the 12 wkPI timepoint. Additionally, at the 12 wkPI timepoint, there was a significant difference (*p* = 0.02, Tukey’s test, BH-corrected) between the SpT animals (0.41 ± 0.05, mean ± SD mm^3^, n = 11) in comparison to the SHAM animal groups (0.36 ± 0.04, mean ± SD mm^3^, n = 3) (Figure 2B).

### 2.6. Hippocampus

For the Hc region volume, there was a significant difference in the longitudinal volume changes measured in SpT animals (*p* = 1.2 × 10^−6^, two-way ANOVA) and VT animals (*p* = 0.006, two-way ANOVA) compared to SHAM control animals. There was no significant difference in longitudinal Hc volume changes between the two control animal groups, i.e., HC and SHAM, or between the two TMEV-induced animal groups.

There was a significant difference for SpT animals at the 20 wkPI timepoint (26.97 ± 1.49 (n = 3) vs. 24.00 ± 0.98 (n = 9), HC vs. SpT, mean ± SD mm^3^) (*p* = 0.005, Tukey’s test, BH-corrected), and at the 28 wkPI timepoint (27.27 ± 0.03 (n = 3) vs. 24.70 ± 0.76 (n = 9), HC vs. SpT mean ± SD mm^3^) (*p* = 0.02, Tukey’s test, BH-corrected). Similarly, for VT animals, there was a significant difference in Hc volume relative to HC animals at the 20 wkPI timepoint (26.97 ± 1.49 (n = 3) vs. 24.93 ± 1.27 (n = 6), HC vs. vehicle, mean ± SD mm^3^) (*p* = 0.01, Tukey’s test, BH-corrected) and at the 28 wkPI timepoint (27.27 ± 0.03 (n = 3) vs. 25.50 ± 1.65 (n = 6), HC vs. vehicle, mean ± SD mm^3^) (*p* = 0.04, Tukey’s test, BH-corrected) (Figure 2E).

### 2.7. Iso-Cortex

For the GM region Cx volume measure, there were no significant differences for longitudinal changes in SpT animals in comparison to VT animals (*p* = 0.368, two-way ANOVA). However, there was a significant difference in the longitudinal changes in the Cx volume measured in SpT animals (*p* = 1.22 × 10^−5^, two-way ANOVA) compared to SHAM control animals. There was no significant difference in longitudinal Cx volume changes between the two control animal groups, i.e., HC and SHAM, or between the two TMEV-induced animal groups. There was a significant difference for SpT animals at the 20 wkPI timepoint (115.92 ± 3.28 (n = 3) vs. 103.72 ± 3.71 (n = 9), HC vs. SpT, mean ± SD mm^3^) (*p* = 9.8 × 10^−4^, Tukey’s test, BH-corrected) and at the 28 wkPI timepoint (116.35 ± 2.48 (n = 3) vs. 103.54 ± 4.28 (n = 9), HC vs. SpT, mean ± SD mm^3^) (*p* = 4.11 × 10^−4^, Tukey’s test, BH-corrected). Similarly, for VT animals, there was a significant difference for Cx volume relative to HC animals at the 20 wkPI timepoint (115.92 ± 3.28 (n = 3) vs. 106.33 ± 6.09 (n = 6), HC vs. vehicle, mean ± SD mm^3^) (*p* = 0.01, Tukey’s test, BH-corrected) and at the 28 wkPI timepoint (116.35 ± 2.48 (n = 3) vs. 107.06 ± 6.45 (n = 6), HC vs. vehicle, mean ± SD mm^3^) (*p* = 0.04, Tukey’s test, BH-corrected) (Figure 2F).

There were similar significant differences between SpT and VT animals compared to SHAM control animals. There was a significant difference in Cx volume for SpT animals at the 20 wkPI timepoint (111.94 ± 2.84 (n = 5) vs. 103.72 ± 3.71 (n = 9), SHAM vs. SpT, mean ± SD mm^3^) (*p* = 0.001, Tukey’s test, BH-corrected) and at the 28 wkPI timepoint (112.87 ± 3.31 (n = 6) vs. 103.54 ± 4.28 (n = 9), SHAM vs. SpT, mean ± SD mm^3^) (*p* = 4.9 × 10^−4^, Tukey’s test, BH-corrected). Similarly, for VT animals, there was a significant difference in Cx volume relative to SHAM animals at the 20 wkPI timepoint (111.94 ± 2.84 (n = 5) vs. 106.33 ± 6.09 (n = 6), SHAM vs. vehicle, mean ± SD mm^3^) (*p* = 0.01, Tukey’s test, BH-corrected) and at the 28 wkPI timepoint (112.87 ± 3.31 (n = 6) vs. 107.06 ± 6.45 (n = 6), SHAM vs. vehicle, mean ± SD mm^3^) (*p* = 0.012, Tukey’s test, BH-corrected) (Figure 2F).

### 2.8. Thalamus

The longitudinal volume changes for the second segmented GM structure Th followed a pattern similar to that of Cx. There were no significant differences for longitudinal changes in ST animals and VT animals (*p* = 0.0914, two-way ANOVA). However, there was a significant difference in the longitudinal changes in Th volume measured in SpT animals (*p* = 4.6 × 10^−4^, two-way ANOVA) compared to SHAM control animals. There was no significant difference for longitudinal Th volume changes between the two control animal groups, i.e., HC and SHAM, or between the two TMEV-induced animal groups. There was a significant difference for SpT animals at the 20 wkPI timepoint (14.95 ± 0.80 (n = 3) vs. 13.69 ± 0.52 (n = 9), HC vs. SpT, mean ± SD mm^3^) (*p* = 0.007, Tukey’s test, BH-corrected) and at the 28 wkPI timepoint (15.03 ± 0.81 (n = 3) vs. 13.62 ± 0.58 (n = 9), HC vs. SpT, mean ± SD mm^3^) (*p* = 0.001, Tukey’s test, BH-corrected). Similarly, for VT animals, there was a significant difference for Th volume relative to HC animals at the 28 wkPI timepoint (15.03 ± 0.81 (n = 3) vs. 13.82 ± 0.48 (n = 6), HC vs. vehicle, mean ± SD mm^3^) (*p* = 0.014, Tukey’s test, BH-corrected) (Figure 2I).

Similarly, there were significant differences between SpT and VT animals compared to the SHAM control group. At the 20 wkPI timepoint, there was a significant difference for SpT animals (14.61 ± 0.34 (n = 5) vs. 13.69 ± 0.52 (n = 9) compared to SHAM vs. SpT, mean ± SD mm^3^) (*p* = 0.0069, Tukey’s test, BH-corrected) and at the 28 wkPI timepoint (14.83 ± 0.58 (n = 6) vs. 13.62 ± 0.58 (n = 9), SHAM vs. SpT, mean ± SD mm^3^) (*p* = 0.0014, Tukey’s test, BH-corrected). Similarly, for VT animals, there was a significant difference in Th volume relative to SHAM animals at the 28 wkPI timepoint (14.83 ± 0.58 (n = 6) vs. 13.82 ± 0.48 (n = 6), SHAM vs. vehicle, mean ± SD mm^3^) (*p* = 0.014, Tukey’s test, BH-corrected) (Figure 2I).

### 2.9. Striatum

Longitudinal volume changes for the structure St followed a pattern similar to that of Cx. There were no significant differences for longitudinal changes in SpT animals in comparison to VT animals (*p* = 0.2695, two-way ANOVA). However, there was a significant difference in the longitudinal changes in St volume measured in SpT animals (*p* = 1.01 × 10^−7^, two-way ANOVA) relative to SHAM control animals. There was no significant difference for longitudinal St volume changes between the two control animal groups, i.e., HC and SHAM, or between the two TMEV-induced animal groups (Figure 2H).

There was a significant difference for SpT animals at the 20 wkPI timepoint (33.56 ± 0.91(n = 3) vs. 30.91 ± 0.89 (n = 9), HC vs. SpT, mean ± SD mm^3^) (*p* = 0.001, Tukey’s test, BH-corrected) and at the 28 wkPI timepoint (33.86 ± 0.68 (n = 3) vs. 30.87 ± 0.91 (n = 9), HC vs. SpT, mean ± SD mm^3^) (*p* = 5.3 × 10^−4^, Tukey’s test, BH-corrected) compared to HC animals. Similarly, for VT animals, there was a significant difference for St volume relative to HC animals at the 20 wkPI timepoint (33.56 ± 0.91 (n = 3) vs. 31.58 ± 1.47 (n = 6), HC vs. VT, mean ± SD mm^3^) (*p* = 0.014, Tukey’s test, BH-corrected) and at the 28 wkPI timepoint (33.86 ± 0.68 (n = 3) vs. 31.66 ± 1.49 (n = 6), HC vs. vehicle, mean ± SD mm^3^) (*p* = 0.008, Tukey’s test, BH-corrected) (Figure 2H).

There were similar significant differences for SpT and VT animals compared to the SHAM control group. There was a significant difference for SpT animals at the 20 wkPI timepoint (33.01 ± 1.38 (n = 5) vs. 30.91 ± 0.89 (n = 9), SHAM vs. SpT, mean ± SD mm^3^) (*p* = 0.001, Tukey’s test, BH-corrected) and at the 28 wkPI timepoint (33.27 ± 1.21 (n = 6) vs. 30.87 ± 0.91 (n = 9), SHAM vs. SpT, mean ± SD mm^3^) (*p* = 0.02, Tukey’s test, BH-corrected) compared to SHAM control animals. Similarly, for VT animals, there was a significant difference in St volume relative to SHAM animals at the 20 wkPI timepoint (33.01 ± 1.38 (n = 3) vs. 31.58 ± 1.47 (n = 6), SHAM vs. VT, mean ± SD mm^3^) (*p* = 0.01, Tukey’s test, BH-corrected) and at the 28 wkPI timepoint (33.27 ± 1.21 (n = 3) vs. 31.66 ± 1.49 (n = 6), SHAM vs. vehicle, mean ± SD mm^3^) (*p* = 0.02, Tukey’s test, BH-corrected) (Figure 2H).

### 2.10. Lateral Ventricles

Interestingly, for longitudinal volume changes for LV, overall, there was a significant difference in average volume measures between SpT and VT animals (*p* = 0.014, two-way ANOVA). Furthermore, there was a significant difference in the longitudinal changes in LV measured in SpT animals (*p* = 7.48 × 10^−8^, two-way ANOVA) compared to SHAM control animals. There was no significant difference for longitudinal LV volume changes between the two control animal groups, i.e., HC and SHAM (Figure 2G).

At the 20 wkPI timepoint, there was a significant difference between the LV volume measured in the SpT animals (5.78 ± 1.01, mean ± SD mm^3^, n = 9) relative to VT (8.76 ± 2.24, mean ± SD mm^3^, n = 6) (*p* = 0.01, Tukey’s test, BH-corrected) and HC control (5.91 ± 1.52, mean ± SD mm^3^, n = 3) (*p* = 0.01, Tukey’s test, BH-corrected) animals. At the 28 wkPI timepoint, there was a significant difference between the LV volume measured in the SpT group (5.90 ± 1.04, mean ± SD mm^3^, n = 9) relative to VT (8.84 ± 2.51, mean ± SD mm^3^, n = 6) (*p* = 0.024, Tukey’s test, BH-corrected) and the SHAM control animals (8.69 ± 2.11, mean ± SD mm^3^, n = 6) (*p* = 0.02, Tukey’s test, BH-corrected). For VT animals, there was a significant difference for LV volume compared to HC animals at the 12 wkPI timepoint (5.53 ± 0.97 (n = 6) vs. 9.09 ± 2.18 (n = 3), HC vs. vehicle, mean ± SD mm^3^) (*p* = 0.014, Tukey’s test, BH-corrected) (Figure 2G). However, the LV measured at 12 wkPI in the SpT group was not significantly different compared to the HC group.

### 2.11. Whole Brain

Taken together, for longitudinal volume changes for WB, there was a significant difference in longitudinal average volume measures between SpT and VT animals (*p* = 0.027, two-way ANOVA). Furthermore, there was a significant difference in the longitudinal changes in LV measured in SpT animals (*p* = 4.74 × 10^−6^, two-way ANOVA) compared to SHAM control animals. There was no significant difference for longitudinal LV volume changes between the two control animal groups, i.e., HC and SHAM (Figure 2A).

### 2.12. Longitudinal Changes in Spine Volume and Lesion Measures throughout the Disease’s Course

Spine MRI images were analyzed to quantify the volume of a specific 9.3 mm spine segment and the lesions within the imaged spine section (Figure 3A,B). There were no significant differences for longitudinal changes in SpT animals compared to VT animals (*p* = 0.725, two-way ANOVA). However, there was a significant longitudinal spine volume change difference between the HC animal group and SpT animals (*p* = 0.002, two-way ANOVA). Additionally, there was a significant difference between VT animals (*p* = 6.6 × 10^−4^, two-way ANOVA) and HC animals. No significant differences were observed between treatment groups at any individual timepoints. At the 28 wkPI scan timepoint, the spine volume for HC animals (21.8 ± 0.72, mean ± SD mm^3^, n = 4) was significantly lower than both VT (24.41 ± 1.03, mean ± SD mm^3^, n = 5) (*p* = 0.01, Tukey’s test, BH-corrected) and SpT animals (24.41 ± 1.18, mean ± SD mm^3^, n = 5) (*p* = 0.013, Tukey’s test, BH-corrected) (Figure 3C).

For spine lesion volumes, there was no significant longitudinal difference between SpT and VT animals (*p* = 0.616, two-way ANOVA). Also, there was no significant difference between HC animals and SpT animals (*p* = 0.12, two-way ANOVA). However, there was a significant difference in average spine lesion volume between VT animals (*p* = 0.01, two-way ANOVA) and HC animals. There were no significant differences between treatment groups at each individual timepoint (Figure 3D).

### 2.13. Microglia-Labeled Cell Density and Phenotype Changes in Response to Treatment

Spine sections were stained for Iba1, and microglial density was quantified manually in GM and WM regions. There was no significant difference in Iba1-labeled cell density between HC, SpT, and VT animals in the spinal cord GM region. The Iba1 cells were categorized into type 1, type 2, and type 3 to denote normal, activated, and amoeboid morphology, respectively, in normal-appearing WM and GM regions.

In the spine GM region, the percentage of type 1 morphology was significantly higher in the HC animals (51.1 ± 3.2, mean ± SD %, n = 3) compared to the VT (30.6 ± 2.59, mean ± SD %, n = 16) (*p* = 0.0324, Tukey’s test, BH-corrected) and SpT animals (32.64 ± 3.81, mean ± SD %, n = 15) (*p* = 0.041, Tukey’s test, BH-corrected) (Figure 4C). Correspondingly, in the GM region, the percentage of type 2 morphology was significantly lower in the HC animals (47.89 ± 3.22, mean ± SD %, n = 3) compared to VT (69.40 ± 2.59, mean ± SD %, n = 16) (*p* = 0.0306, Tukey’s test, BH-corrected) and SpT animals (67.36 ± 3.81, mean ± SD %, n = 16) (*p* = 0.0398, Tukey’s test, BH-corrected) (Figure 4C). For the analyzed spine tissue stain images, no amoeboid-appearing microglial cells were found in the normal-appearing GM.

In the spine WM region, the percentage of type 1 morphology was significantly higher in the HC animals (63.13 ± 4.07, mean ± SD %, n = 3) compared to VT (36.46 ± 3.32, mean ± SD %, n = 16) (*p* = 0.0256, Tukey’s test, BH-corrected) and SpT animals (38.64 ± 4.27, mean ± SD %, n = 16) (*p* = 0.029, Tukey’s test, BH-corrected) (Figure 4C). Correspondingly, in the spine WM region, the percentage of type 2 morphology was significantly lower in the HC animals (36.87 ± 4.07, mean ± SD %, n = 3) compared to VT (62.80 ± 3.10, mean ± SD %, n = 16) (*p* = 0.0266, Tukey’s test, BH-corrected) and SpT animals (60.27 ± 4.03, mean ± SD %, n = 16) (*p* = 0.0256, Tukey’s test, BH-corrected) (Figure 4C). For the spine tissue analyzed, no amoeboid-appearing microglial cells were found in the normal-appearing WM for HC animals. However, in the WM region, a small fraction of the Iba1-labeled cells were detected in the VT (0.74 ± 0.74, mean ± SD %, n = 16) and SpT (1.55 ± 0.64, mean ± SD %, n = 16) animals.

## 3. Discussion

### 3.1. SpT TMEV Animals Presented an Increased Disease Severity

In agreement with our previous findings [15], the clinical measures in this study showed that TMEV-induced animals exhibited disease progression over a 7-month time frame. The primary hypothesis for this study was that Sp would reduce the progression of TMEV-induced clinical disability and decline in the motor skills, as measured by rotarod testing [10]. For example, in the EAE disease model, Sp was shown to reduce clinical disability in the treated animals and improve other outcome measures, such as myelin integrity [9]. However, in the present study, the CDS, BW, and rotarod retention time results did not align with the primary hypothesis. There was an increase in disease severity in the SpT TMEV animals. These contradictory results may be attributed to the virus-driven pathology in TMEV disease. In the TMEV model, T cells not only contribute to inflammatory pathology, but also help constrain the increase in viral load during the acute phase, thereby mitigating disease severity. Conversely, the pathology in EAE disease primarily stems from T cell-driven autoimmune responses [16]. Therefore, unlike in EAE studies, Sp treatment in the TMEV model might inadvertently augment disease-driven inflammatory activity by suppressing the immune T cell response needed to limit viral load in TMEV-infected animals.

### 3.2. SpT Has Limited Impact on Brain Region-Specific Volume Changes

In this study, we observed a decrease in the volumes of several brain region segments in the TMEV-induced animals in comparison to HC or SHAM control animals. For example, there was a significant longitudinal decline in the Cx, Th, and St volume in the TMEV-induced VT animals relative to HC animals and SHAM control animals at the chronic disease phase. Sp treatment did not affect these TMEV-induced volume decreases. There was no significant difference in longitudinal changes between VT and SpT animals for most of the brain regions evaluated in this study.

TMEV-induced LV increase relative to control animals was observed at earlier acute disease stages. This increase in LV volume was noted in SHAM control animals as well, suggestive of trauma-induced inflammation due to IC injection SHAM surgery [15]. However, in contrast to brain region-specific volume decreases, there was a significant difference in average longitudinal LV volume measure changes between SpT and VT animals.

TMEV viral infection induces encephalomyelitis, particularly during the acute phase of the disease. This is triggered by T cell-mediated cellular activity, aimed at eliminating viral load within the infected animal’s brain, resulting in increased LV volume [17]. Sp treatment and possible suppression of lymphocyte transport started 1 month post-induction. Such suppression of early immune cell response would explain alterations to longitudinal LV volume trace in SpT TMEV mice in comparison to VT animals. Limiting the baseline clearance of TMEV and, consequently, resulting in a relatively severe inflammation in SpT animals in the chronic disease phase would also explain severe disease progression measured using CDS, BW, and rotarod retention time in the SpT group.

### 3.3. Effect of Sp Treatment on TMEV-Induced Spine Volume Changes

Hind limb paralysis was evaluated to assign a CDS score and to measure disease progression in TMEV mice. Neurodegeneration in the spine is responsible for the presented hind limb paralysis and loss of motor skills observed in TMEV mice [13]. Hence, in this study, the effect of Sp treatment on spine pathology was evaluated using spine MRI imaging. Spine MRI images were analyzed to quantify the volume of a specific 9.25 mm spine segment and to identify lesions within the imaged spine section. The results showed no significant differences in longitudinal changes between SpT and VT animals. HC animals exhibited significantly lower spine volumes compared to both SpT and VT animals, indicating that TMEV-induced inflammation leads to increased spine volume. These changes in spine volume align closely with recently published findings on EAE disease-induced spinal cord volume changes, where Althobity et al. reported a significant increase in WM spinal cord volume and a slight reduction in GM spinal cord volume in EAE mice compared to healthy control mice [18].

### 3.4. Spine Histology

In addition to neuroimaging of the spine, the present study investigated the effect of Sp treatment on microglial morphology in GM and WM regions of the spinal cord in TMEV mice. Microglial cells were identified based on their Iba1 staining pattern and were then morphologically categorized. The analysis revealed no significant difference in microglial density between HC, SpT, and VT animals in the spinal cord GM region. However, in both the GM and WM regions, the percentage of type 1 morphology was significantly higher in the HC animals compared to both SpT and VT animals, while the percentage of type 2 morphology was significantly lower in the HC animals compared to VT and SpT animals. These findings suggest that, in spite of severe clinical outcomes, Sp treatment does not significantly affect microglial density or activation in the spinal cord. The impact of Sp on microglia may be confounded by TMEV infection, which possibly induces detrimental acute disease-phase lymphocyte transport and anti-inflammatory response during the chronic phase of the disease. Hence, further studies investigating the impact of Sp treatment at later disease stages and on different TMEV viral loads may be needed to better discern the effect of Sp treatment on the TMEV disease model.

## 4. Materials and Methods

### 4.1. Subjects and Study Design

All animal procedures were conducted as per institutional animal care and use committee-approved protocols. In all, 102 4–5-week-old SJL/J mice, ordered from Jackson Laboratories, were allowed to acclimate to the environment for 1 week. TMEV infection was inducted at 6–8 weeks of age. One-month post TMEV induction, 44 were treated with Sp (Sp-treated, SpT), 44 were treated with placebo (vehicle-treated, VT). Furthermore, six animals served as sham surgery controls (SHAM) and eight were used as age-matched healthy controls (HC). Clinical disability score (CDS), body weight (BW), and rotarod retention time measures were quantified for all 102 animals. Then, 58 out of the 102 mice underwent scanning and histological analysis. These 58 animals were divided into 2 scanning groups as follows: Group 1 (SpT = 15 mice, VT = 15 mice, SHAM = 6, and HC = 5 mice) underwent brain imaging; Group 2 (SpT = 7 mice, VT = 7 mice, and HC = 3 mice) underwent spinal cord imaging. In the brain volume scanning group, four VT animals and one HC animal did not recover from anesthesia during the first timepoint scan. One SpT animal from the spine volume scanning group animal did not recover from anesthesia during their first MRI scan timepoint. All scans were conducted at 1, 3, 5, and 7 months post-TMEV induction, except for the HC animals in the spine scan group, which were scanned at 1 and 7 months post-TMEV induction timepoint (Appendix A).

### 4.2. Induction of TMEV Infection

The virus was generated as previously described [19] and was delivered into the CNS by bilateral intracerebral injection at 6–8 weeks of age. The SJL/J strain was used because of its genetic susceptibility to acquiring a chronic, demyelinating disease from TMEV [12]. A tuberculin needle was used to administer 3 × 10^6^ plaque-forming units of TMEV into the cerebrum by injection through the skull, past the meningeal layer in isoflurane-anesthetized mice. This was performed by piercing a thin layer of skull with the needle, approximately 1.5 mm posterior to bregma and 1.5 mm away from midline, and inserting the needle just past the bevel. A William’s Collar, or a tube partially sheathing the needle, was used to restrict needle depth to 2 mm.

To verify successful inoculation of TMEV virus in IC-injected mice, we assessed the anti-TMEV immune response. We performed ELISA tests on peripheral blood at 2 months post-TMEV IC injection, drawn from the facial vein, to detect the presence of anti-TMEV antibodies. These tests utilized a TMEV antigen ELISA kit (XpressBio, Thrumont, MD, USA). All the animals that were IC-injected with TMEV were confirmed to have TMEV-reactive blood serum [20].

### 4.3. Treatment

One month after TMEV injection, treatment with Sp (Novartis Pharma AG, Basel, Switzerland) or VT placebo was administered daily using oral gavage. Sp was given to 22 mice at a dosage of 3 mg/kg in 0.6% carboxymethylcellulose/0.5%. An equivalent volume of identical vehicle solution without Sp was given to 22 mice using oral gavage. Treatment commenced one month post-TMEV induction, coinciding with the transition from acute onset to the chronic phase of TMEV disease progression. The therapy was sustained until euthanasia was performed on the animals for ensuing analysis [21]. The additional sham surgery cohort of 6 saline-injected mice received no treatment. Finally, 8 mice in the age-matched HC group did not receive any IC injection or treatment, to statistically isolate IC injection-related effects.

### 4.4. Clinical Scoring

A previously published clinical disability score system was used [21]. The disability was rated on a modified 4-point scale: 0—no gait abnormality, 1—mild waddling gait, 2—severe waddling gait, 3—severe waddling gait with impaired righting due to extensor spasm, and 4—moribund [15].

### 4.5. Rotarod

Mice were evaluated for motor disability by rotarod assay at each timepoint. The rotarod began at one rotation per minute (rpm) and accelerated to 70 rpm with the score quantified as the number of seconds the mouse remains on the rotarod. Up to 3 tests were administered at each timepoint. The best test score at each timepoint was considered for final data analysis. Mice were trained on the rotarod two times for three days prior to the baseline timepoint. Training on the rotarod consisted of a constant speed of 4 rpm for 5 min on the first day, then accelerating the task on the second and third days [22].

### 4.6. MRI Scanning

Mice were anesthetized with 4–5% isoflurane and then maintained with 1–3%. MRI scanning was conducted on a horizontal 9.4 T small animal MRI scanner (Bruker Biospin, Biospec 94/20 USR, Billerica, MA, USA) with a cryogenically cooled surface transmit–receive coil placed over the head of the mouse. The MRI scan protocol lasted approximately one hour. In order to prevent hypothermia, both the probe head and the bed were heated to 37 °C. In order to prevent dehydration, mice were injected with 1 mL saline subcutaneously after the imaging procedure. Lubricant was placed on the eyes in order to prevent drying. Respirations were monitored, and isoflurane was adjusted accordingly to maintain 20–50 breaths per minute (which is the optimal range for TMEV animals) [22].

### 4.7. Brain Volume Determination

Automated brain region-specific segmentation was performed for each brain scan as previously described [13]. Briefly, an ultra-high-resolution 3D multi-echo gradient-echo (MEGRE) sequence was used for volumetric analysis (2.38 ms 1st echo, 4.4 ms spacing, 9 mono-polar echoes, 90 ms repetition time, 18° flip angle, 75% partial-Fourier in read direction, 27 × 14 × 8mm field of view, 252 × 180 × 100 matrix, 80 μm isotropic resolution, 27 min acquisition time) [17].

In order to segment dozens of brains efficiently, a computer-automated template and atlas-driven segmentation procedure was created [13,23,24]. To generate the atlases, regions of interest (ROIs) were outlined manually using 3D Slicer (www.slicer.org (accessed on 10 May 2022) in 10 scans from 3 different representative animals [24], with the visual aid of Waxholm Atlas v0.6.2.32. In addition to whole brain volume (Wb), ROIs included the anterior commissure (AC), cerebellum (Cb), corpus callosum (CC), hippocampus (Hc), thalamus (Th), iso-cortex (Cx), lateral ventricles (LV), and striatum (St). The cortex encompassed the entire bilateral cortical layer; the basal ganglia encompassed the bilateral caudoputamen, bilateral nucleus accumbens, and bilateral globus pallidus; the Th encompassed the bilateral Th; and the ventricular space encompassed bilaterally parenchyma-enclosed ventricular areas (Appendix A).

To segment the individual scans, ANTs (Advanced Normalization Tools) were used to generate a transformation matrix for each scan to non-linearly align the templates to the target image [25]. Subsequently, the transformation matrix was utilized to align the labeled atlases associated with the templates to the target space. The selection of the appropriate structure label at each voxel was accomplished through a joint fusion-weighted voting technique, considering the local voxel-wise correlations with the corresponding templates. This approach favored the selection of atlas labels from templates that exhibited better matching within each region, resulting in the final choice. This approach favored the selection of atlas labels from templates that exhibited better matching within each region, resulting in the final choice [15].

### 4.8. Spinal Cord Lesion and Volumetric Evaluation

Spinal cord imaging was performed with a dedicated mouse spine coil using a 3D T2-TurboRARE sequence (TR 400 ms, TE 21.46 ms, FOV 20 × 10 × 40 mm, matrix 192 × 96 × 192, RARE factor 8, 2 averages, 30 min acquisition time). A one-minute high-resolution MGRE localizer was used to place the imaging slab. All spinal cord lesion and atrophy measures were performed by a single analyst who was blinded to the disease status. Volumes were calculated by applying a semi-automated iso-contouring technique using JIM software 6.0 (Xinapse Systems, Leicester, UK. http://www.xinapse.com, (accessed on 10 May 2022)). For consistency, the image prescription was always placed (to start) at the T10 vertebrae to image the same 9.35 mm long section of the spine, for all timepoints and for each animal.

### 4.9. Histological Analysis

Animals underwent cardiac perfusion with saline, followed by 4 percent paraformaldehyde post-euthanasia for tissue extraction and preservation. Cryopreserved brains were cryosectioned into 16 µm thick sections to collect two sections to sample from each region of interest. Tissue sections were stained for microglia and macrophage marker Iba1 (Wako Chemicals, Osaka, Japan) [26]. The Iba1-stained cells were classified into three types based on morphology. Type one denoted inactive microglia with ramified morphology, and activated microglia with amoeboid morphology [27] were quantified within the GM and white matter (WM) regions in spine sections. Open-source software programs ImageJ and Fiji were used for analysis of immune-fluorescently stained brain section images [24,28].

### 4.10. Statistical Analysis

Individual MRI results were compared back to the baseline by paired *t*-tests and across the treatment conditions by unpaired *t*-tests. Comparison between groups was performed using linear mixed-effect modeling. Treatment-wise comparison *p* values were adjusted with the Benjamin–Hochberg correction and, for pairwise comparisons between timepoints within each treatment, *p* values were adjusted with Tukey’ to correct for multiple testing. Study data analyses and graphing were completed using R programming, using ggplot, tidyr, lmer_test, and lsmeans library files. Longitudinal clinical and MRI data were analyzed using linear mixed-modeling, with animal IDs as random effects and time and treatment as fixed effects. The F-test results from linear mixed-models were corrected for multiple testing using the Bonferroni–Hochberg correction [13,22].

## 5. Conclusions

In conclusion, the results of this study suggest that Sp treatment had limited neuroprotective effects in the CNS of TMEV-induced demyelinating disease mice. The study also found no adverse effects of Sp treatment on spine pathology, with no significant differences for longitudinal changes in Sp animals compared to VT animals. However, Sp treatment suppressed the TMEV infection-induced inflammation in the LV. Similarly, analysis of spine tissue sections showed no significant difference in microglial density between HC, SpT, and VT animals in the spinal cord GM region, suggesting that Sp treatment did not cause significant inflammation in the spinal cord. The worsening of the clinical measures would suggest that Sp’s immune-suppressing mechanisms of action dominated the treatment’s neuroprotective mechanisms of action. Future experiments may focus on altering experimental design to include Sp treatment that starts at a later timepoint, to study the impact of Sp treatment on TMEV’s chronic phase inflammatory pathology and varying TMEV viral infection dosages.

## Figures and Tables

**Figure 1 ijms-24-12990-f001:**
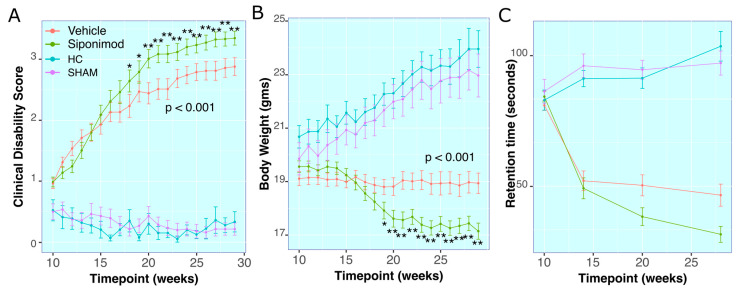
Effect of siponimod treatment on clinical course of TMEV disease: (**A**) Clinical disability score (CDS) for all animals throughout the study. (**B**) Body weights for all animals throughout the study. (**C**) Rotarod retention time in seconds for all animals throughout the study. The red line represents animals treated with siponimod at 3 mg/kg, the olive-green line represents vehicle-treated, the blue line represents healthy controls, and the purple line represents saline intracerebral SHAM injection. The color assignments for the different treatments are outlined in the figure. The two-way ANOVA calculated *p* value for siponimod vs. vehicle comparison are presented as text on the graph. * and ** represent *p* < 0.05 and *p* < 0.01, respectively, for Bonferroni–Hochberg-corrected *p* value for individual timepoint comparison for siponimod vs. vehicle-treated animals. The distance between the two whiskers represents inter-quartile distance from first to third quartile.

**Figure 2 ijms-24-12990-f002:**
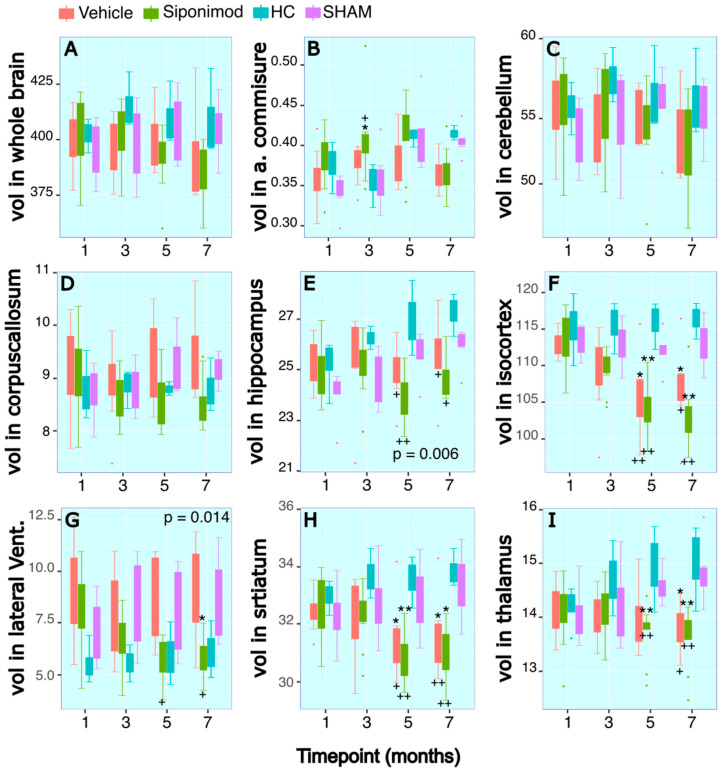
Longitudinal tracing of brain region-specific volumetric changes: (**A**–**I**) Longitudinal changes in volume (mm^3^) in the regions of whole brain, anterior commissure, cerebellum, corpus callosum, hippocampus, iso-cortex, lateral ventricles, striatum, and thalamus, respectively. The x axis presents scan timepoint in months and the y axis presents volume measured in mm^3^. The red graphs represents animals treated with siponimod at 3 mg/kg, the olive-green line represents vehicle-treated, the blue line represents healthy controls, and the purple graphs represents saline intracerebral SHAM injection. The color assignments for the different treatments are outlined in the figure. The two-way ANOVA-calculated *p* value for siponimod vs. vehicle-treated comparisons are presented as text on the graph. * and ** represent *p* < 0.05 and *p* < 0.01, respectively, for Bonferroni–Hochberg-corrected *p* value for individual timepoint comparison for siponimod vs. SHAM control group. + and ++ represent *p* < 0.05 and *p* < 0.01, respectively, for Bonferroni–Hochberg-corrected *p* value for individual timepoint comparisons for siponimod vs. healthy control animal group. The distance between the two whiskers represents inter-quartile distance from first to third quartile.

**Figure 3 ijms-24-12990-f003:**
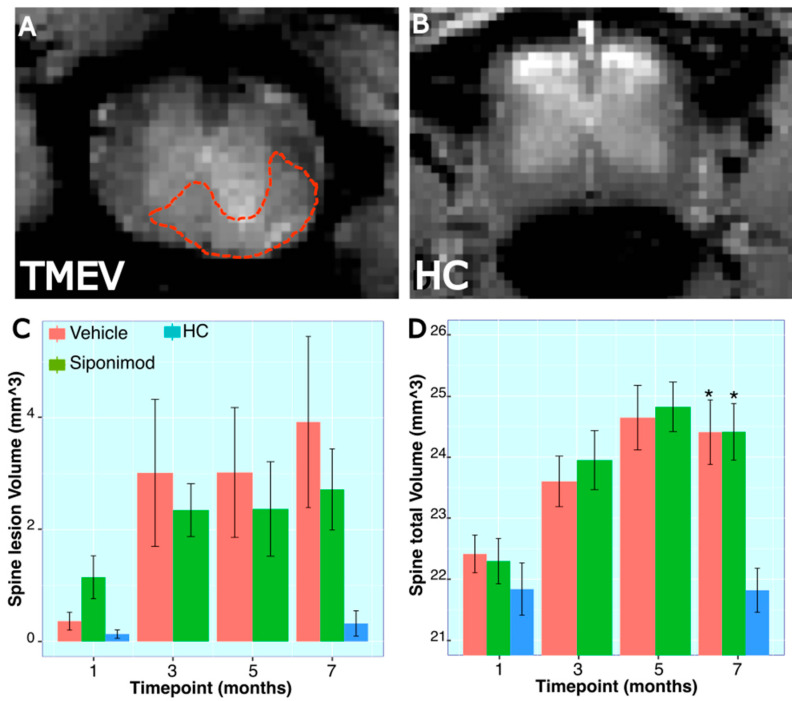
Longitudinal trace for spinal volume and lesion measures in TMEV mice: (**A**,**B**) Representative MRI spine scan images from a TMEV and a healthy control animal, respectively. The red dotted line shows an example of a manually drawn lesion ROI on the spine. (**C**,**D**) Spine lesion volume measured in a 9.35 mm long section of the spine, starting at the T10 vertebral disc. * represent *p* < 0.05, for Bonferroni–Hochberg-corrected *p* value for individual timepoint comparison with HC animal data. The red bar represents animals treated with siponimod at 3 mg/kg, the olive-green bar represents vehicle-treated, and the blue bar represents healthy controls. The color assignments for the different treatments are outlined in the figure. The distance between the two whiskers represents inter-quartile distance from first to third quartile.

**Figure 4 ijms-24-12990-f004:**
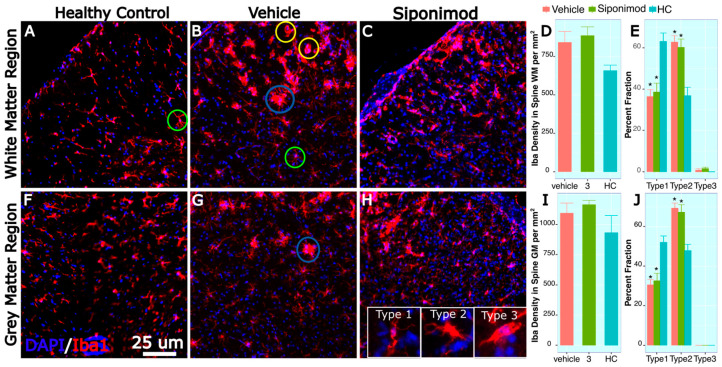
Spine microglial density changes in siponimod-treated TMEV animals: Spine tissue sections were stained for Iba1 and DAPI to label microglial cells and all cell nuclei, respectively. Images (**A**–**C**) are representative images of the spine white matter region, collected from healthy control, vehicle-treated, and siponimod-treated animals, respectively. (**F**–**H**) are representative images of the spine gray matter region collected from healthy control, vehicle-treated, and siponimod-treated animals, respectively. (**D**,**I**) are graphs presenting the density of Iba1-labeled cells in white matter and gray matter, respectively. (**E**,**J**) are graphs presenting percent fraction of the Iba1-labeled cells presenting type 1, 2, and 3 morphology. The inset in (**H**) contains representative images for normal morphology (type 1), activated branched morphology (type 2), and (type 3) amoeboid morphology. Green, blue, and yellow circles identify Iba1-labeled cells presenting type 1, 2, and 3 morphology, respectively, in (**A**–**H**). * *p* < 0.05, for Bonferroni–Hochberg-corrected *p* value for individual timepoint comparison with HC animal data. The red bar represents animals treated with siponimod at 3 mg/kg, the olive-green bar represents vehicle-treated, and the blue bar represents healthy controls. The color assignments for the different treatments are outlined in the figure. The scale bar refers to 25 μm length. The distance between the two whiskers represents inter-quartile distance from first to third quartile.

## Data Availability

Not applicable.

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
