# Peer review of "Effect of Siponimod on Brain and Spinal Cord Imaging Markers of Neurodegeneration in the Theiler’s Murine Encephalomyelitis Virus Model of Demyelination"

_ijms, 2023, doi:10.3390/ijms241612990_

Round 1

Reviewer 1 Report

The manuscript by Pol et al. examines the effects of siponimod neurodegeneration driven by Thieler's murine encephalomyelitis virus model of MS. The authors have performed a well-designed examining the effects of siponimod on volumes of various brain regions and neuroinflammation in spinal cord as assesed by Iba1 immunoreactivity.

Even though the study has some potentials, serious limitations and concers must be addressed:

Introduction

Overall, the introduction is diffuse and does not provide a clear background as to what and why the authors undertook this study. The sections on human pathology should be shortened and the focus should be more on TMEV as a model. In addition, it is not clear why TMEV was chosen to study the effect of siponimod, which is already used for MS. The authors need to emphasise the novelty of studying siponimod on TMEV.

For example, the reviewer could not understand whether the authors intended to investigate the effects of siponimod on glia-driven neurodegeneration or on autoimmune cell-driven inflammation/demyelination. In this regard, there are different statements in the abstract and the introduction. Finally, both the acute and chronic phases of TMEV infection are largely driven by CD4 and CD8 immune cells (please see PMID 14726460), so it remains unclear how the authors could separate the effects of siponimod on glial cells from the effects on CD4/CD8 lymphocytes, which is a major limitation of the study. 

Material and Methods

The study design and statistics in the study are correctly chosen and number of animals is high enough to discover even small biological effects of siponimod. However, some methods are poorly described which prevents the independent reproduction of the study.

1) The authors should explicitly state what are the stereotaxic coordinates where the virus was injected, at what region and why that region is chosen? Are there any differences in the pathology if the virus is injected in e.g. cortex vs hippocampus. What is the volume in which the virus is injected and did the authors exclude the effects of stab wound injury?

2) It is not clear how the authors performed ImageJ analysis of Iba1. This must be improved.

3) TMEV induction validation should be included in the suplementary material

Results

Authors should emphasize more clearly why they chosen time point for staring the treatment. It should be clearly stated for how long the treatment was applied. Did the authors consider the effect of prolonged oral gavage as an important stress factor? 

It should be easier to follow Figures if the legend for each group was embedded at the picture, not only as a text in Figure legends. 

There is no explanation for + sign, only for *

Why authors chose only Iba1 to asses the changes in glial cells? What about astrocytes and oligodendrocytes which are also important depos of virus replication.

Authors must perform immunohistochemistry on CD4/CD8 lymphocytes to observe how siponimod affects the infiltration of these cells in the brain/spinal cord tissue.

Discussion

The authors should more elaborate on somewhat contradictory results on effects of siponimod and CD4 and CD8 driven inflammation. Authors should clearly state what is the significance of their data regarding the effects of siponimod and what is the rationale for TMEV model compared to other models.

English language needs modest improvement. 

Author Response

Reviewer 1:

The manuscript by Pol et al. examines the effects of siponimod neurodegeneration driven by Thieler's murine encephalomyelitis virus model of MS. The authors have performed a well-designed examining the effects of siponimod on volumes of various brain regions and neuroinflammation in spinal cord as assesed by Iba1 immunoreactivity.

Even though the study has some potentials, serious limitations and concerns must be addressed:

Introduction

Overall, the introduction is diffuse and does not provide a clear background as to what and why the authors undertook this study. The sections on human pathology should be shortened and the focus should be more on TMEV as a model. In addition, it is not clear why TMEV was chosen to study the effect of siponimod, which is already used for MS. The authors need to emphasise the novelty of studying siponimod on TMEV.

The reviewer’s comments are well taken. We have revised the introduction section to address comments made by the reviewer. The sections in the introduction with lower relevance to the study have been truncated.  Emphasis has been placed on the rational for adopting TMEV model for this study.

For example, the reviewer could not understand whether the authors intended to investigate the effects of siponimod on glia-driven neurodegeneration or on autoimmune cell-driven inflammation/demyelination. In this regard, there are different statements in the abstract and the introduction.

The study aimed to investigate chronic-inflammation driven neurodegeneration that is closely modelled in TMEV disease model. With the edits made to the introduction section, we aim to make this more-clear in the manuscript.

Finally, both the acute and chronic phases of TMEV infection are largely driven by CD4 and CD8 immune cells (please see PMID 14726460), so it remains unclear how the authors could separate the effects of siponimod on glial cells from the effects on CD4/CD8 lymphocytes, which is a major limitation of the study. 

It is accurate to say that the pathology in TMEV is a complex interplay of CD4/CD8 and glial cells driven pathological mechanisms. However, in contrast to EAE disease model, the chronic phase pathology is not driven by a directly induced auto-immune response. TMEV pathology is driven by immune cell induced inflammatory response to the viral load. We aimed to investigate this in the presented study.

Material and Methods

The study design and statistics in the study are correctly chosen and number of animals is high enough to discover even small biological effects of siponimod. However, some methods are poorly described which prevents the independent reproduction of the study.

  • The authors should explicitly state what are the stereotaxic coordinates where the virus was injected, at what region and why that region is chosen? Are there any differences in the pathology if the virus is injected in e.g. cortex vs hippocampus. What is the volume in which the virus is injected and did the authors exclude the effects of stab wound injury?

We thank the reviewers for this feedback. The study adopted the protocol first described by Lipton et al 1975 (PMID 164412) which uses free hand injection without using a stereotax. The adopted protocol ensures a consistent depth of injection for all the animals using a Williams collar on the syringe needle. It is positioned approximately 1.5 mm posterior to bregma and 1.5 mm from the midline to target the cortical region to inject 30 ul of virus solution. The virus has been shown to spread to cerebellum and spine within 10-20 days of inoculation. Therefore, stereotactic injection’s sub millimeter accuracy should not effect the long term outcomes of the injection. The MRI scans were conducted a few months after the TMEV injection. To account for the impact of IC injection scar tissue, a SHAM surgery group was included in the study. Therefore, the minimal scar tissue did not significantly affect the final volume outcome measures. The method’s section was edited to include these details.

  • It is not clear how the authors performed ImageJ analysis of Iba1. This must be improved.

The Iba1 stained cell morphology was assessed manually by a single rater. After identifying a Iba1 stained cell, based on Wu et al, (Pubmed ID 16155637) labelled cells were categorized in to the three subtypes (normal (type 1), ramified activated (type 2) and ameboid (type 3)) to trace the activation of microglia in response to TMEV disease progression.  The methods section, figure and figure legends have been edited to better describe Iba1 cell analysis.

  • TMEV induction validation should be included in the supplementary material

We thank the reviewers for this suggestion. The induction of TMEV is validated using ELISA assay to detect anti-TMEV anti bodies in peripheral blood circulation. The methods section has been edited present this information. The detection kit includes a mouse derived positive control sample and a plate with positive and negative antigen wells. The kit does not include a tittered antibody concentration standard to determine the exact antibody concentration. Hence the kit only confirms if the animal sera positively reacts to TMEV antigens without determining the exact concentration.

Results

Authors should emphasize more clearly why they chosen time point for staring the treatment. It should be clearly stated for how long the treatment was applied.

We apologize for this oversight. The methods section has been updated to clearly stipulate the TMEV treatment timepoints. The treatment was initiated at 1 month post induction. The bi-phasic TMEV disease is divided into acute (0-2 months post induction) and chronic inflammatory phase (after 2 months post induction).   To target the transition into the chronic phase of the disease.

Did the authors consider the effect of prolonged oral gavage as an important stress factor? 

Oral gavaging was factored into our experiment design. To control for this vehicle animals were administered placebo as oral gavage as well. This information has been added to the methods section.

It should be easier to follow Figures if the legend for each group was embedded at the picture, not only as a text in Figure legend. The legends have been updated to include this detail.

The reviewer’s point is well taken. If the journal publication guidelines allow for it, we have submitted edited figures with color legends included in each.

There is no explanation for + sign, only for *

We have updated the figure legends to explain + and ++

Why authors chose only Iba1 to asses the changes in glial cells? What about astrocytes and oligodendrocytes which are also important depos of virus replication.

Histological assessment of astrocytes and oligodendrocyte will be explored in the future studies.

Authors must perform immunohistochemistry on CD4/CD8 lymphocytes to observe how siponimod affects the infiltration of these cells in the brain/spinal cord tissue.

We aimed to investigate the inflammatory status within the tissue of interest in response to Siponimod treatment. We used microglial activation as a proxy to evaluate this. Investigating the status oligodendrocyte differentiation, and astrocyte density will further illustrate effect of siponimod treatment on glia and would be beyond the scope of this study. In future studies investigating the underlying mechanisms of our observations, we aim to characterize the oligodendroglial and astroglial population.

Discussion

The authors should more elaborate on somewhat contradictory results on effects of siponimod and CD4 and CD8 driven inflammation. Authors should clearly state what is the significance of their data regarding the effects of siponimod and

The discussion section has been revised to elucidate the divergent roles of T cells in EAE and TMEV pathologies. The initial paragraph in the discussion posits that siponimod treatment, by suppressing T cells—which are crucial in controlling the viral load in TMEV infection—may exacerbate the severity of the disease, as indicated by disability and other clinical measurements. Our future research plans include modifications to the treatment regimen to potentially mitigate the adverse effects resulting from T cell suppression in the TMEV model.

what is the rationale for TMEV model compared to other models.

We have revised the introduction section to address some of the points made by the reviewer. The sections in the introduction with lower relevance to the study have been truncated.  Emphasis has been placed on the rational for adopting TMEV model for this study. 

Comments on the Quality of English Language.

English language needs modest improvement. 

We have improved the quality of English language by proofreading a manuscript by an expert.

Reviewer 2 Report

Inthe presented study, the authors investigate the possible effects of S1P modulator on a mouse model of inflkammatory CNS diseasdes mimicking MS. To do so they used a Theiler's murine encephalomyelitis virus (TMEV) mous model of MS; mice neurodegeneration mechanisms were investigated by longitudinal analyses with 9.4 T MRI and by histological analyses of  grey and white matter spine sections.

I have some suggestion that need to ae addressed:  

1) Introduction should be shortened, particularly in the paragraph referring to phase 3 trila Expand; this does not add significant clues to the study rationale.

2) On the opposite the rationale for using this specific mouse model as a model for MS associated neurodegeneration is not clear to me and need to be discussed in detail particularly considering existing literature about putative mechanisms of prevention from neural and glial degeneration in humans by S1P treatments.

3) Histological analyse should be further detailed.

Author Response

Reviewer 2:

In the presented study, the authors investigate the possible effects of S1P modulator on a mouse model of inflkammatory CNS diseasdes mimicking MS. To do so they used a Theiler's murine encephalomyelitis virus (TMEV) mous model of MS; mice neurodegeneration mechanisms were investigated by longitudinal analyses with 9.4 T MRI and by histological analyses of  grey and white matter spine sections.

I have some suggestion that need to ae addressed:  

  • Introduction should be shortened, particularly in the paragraph referring to phase 3 trila Expand; this does not add significant clues to the study rationale.

We have shortened the introduction section to remove distantly related information.

  • On the opposite the rationale for using this specific mouse model as a model for MS associated neurodegeneration is not clear to me and need to be discussed in detail particularly considering existing literature about putative mechanisms of prevention from neural and glial degeneration in humans by S1P treatments.

The introduction section has been revised to describe the rationale behind using TMEV disease model.

3) Histological analyses should be further detailed.

We have revised sections of histological analyses.

Round 2

Reviewer 1 Report

The Authors have thoroughly addressed all the comments and revised the manuscript accordingly